# Psychological distress among health service providers during COVID-19 pandemic in Nepal

Khagendra Kafle[1], Dhan Bahadur Shrestha[2]*, Abinash Baniya[3],
Sandesh Lamichhane[3], Manoj Shahi[3], Bipana Gurung[3], Partiksha Tandan[3],
Amrita Ghimire[4], Pravash Budhathoki[5]

1 Department of Psychiatry, Chitwan Medical College Teaching Hospital (CMCTH), Chitwan, Nepal,
2 Mangalbare Hospital, Morang, Nepal, 3 Chitwan Medical College Teaching Hospital (CMCTH), Chitwan,
Nepal, 4 Department of Psychiatric Nursing, Chitwan Medical College Teaching Hospital, Chitwan, Nepal,
5 Dr Iwamura Memorial Hospital, Bhaktapur, Nepal

* medhan75@gmail.com

## Abstract

### Background

COVID-19 pandemic has provoked a wide variety of psychological problems such as anxiety, depression, and panic disorders, especially among health service providers. Due to a greater risk of exposure to the virus, increased working hours, and fear of infecting their families, health service providers are more vulnerable to emotional distress than the general population during this pandemic. This online survey attempts to assess the psychological impact of COVID-19 and its associated variables among healthcare workers in Nepal.

### Materials and methods

For data collection purposes, Covid-19 Peritraumatic Distress Index (CPDI) Questionnaire, was used whose content validity was verified by Shanghai mental health center. Data for the survey were collected from 11 to 24 October 2020 which was extracted to Microsoft Excel-13 and analyzed.

### Results

A total of 254 health care workers from different provinces of the country participated in this study with a mean age of 26.01(± 4.46) years. A majority 46.9% (n = 119) of the participants were not distressed (score ≤28) while 46.5% (n = 118) were mild to moderately distressed (score >28 to ≤51) and 6.7% (n = 17) were severely distressed (score ≥52) due to the current COVID-19 pandemic. Female participants (p = 0.004) and participants who were doctors by profession (p = 0.001) experienced significantly more distress.

### Conclusions

COVID-19 pandemic has heightened the psychological distress amongst health care service providers. The findings from the present study may highlight the need for constructing and implementing appropriate plans and policies by relevant stakeholders that will help to

**Data Availability Statement:** All relevant data are within the manuscript and its Supporting information files.

**Funding:** The author(s) received no specific funding for this work.

**Competing interests:** The authors have declared that no competing interests exist.

mitigate the distress among health service providers in the current pandemic so that we can have an efficient frontline health workforce to tackle this worse situation.

## Introduction

The Coronavirus disease 2019 (COVID-19), as named by the World Health Organization (WHO), first emerged as a cluster of unknown pneumonia cases in Wuhan in late December 2019 [1, 2]. This outbreak had spread substantially throughout the world for which it was declared as a Public Health Emergency of International Concern (PHEIC) on 30th January 2020 and as a pandemic by the World Health Organization (WHO) on March 11, 2020 [3, 4]. As of October 29, 2020, COVID-19 has accounted for 43,766,712 confirmed cases and 1,163,459 deaths across 219 territories [5]. Nepal registered its first case of COVID-19 on January 23, 2020. Despite adopting operative measures like nationwide lockdown, social distancing, and travel restrictions, the COVID-19 cases are in increasing trend in Nepal. Till October 29, 2020, there have been 164,718 confirmed cases of COVID-19, of which 124,862 (75.8%) had recovered and 904 (0.6%) deaths have been recorded [6].

The current COVID-19 pandemic has not only caused significant threats to people's physical health and lives but has also provoked a wide variety of psychological problems such as anxiety, depression, and panic disorders [7]. During acute health crises like the current COVID-19 pandemic, healthcare systems and facilities are under extreme pressure for providing appropriate diagnostic and treatment services due to which the working life of health service providers in affected regions has become more stressful than normal [8]. Health service providers who are working as front liners in the current pandemic are more vulnerable to emotional distress than the general population as they have a greater risk of exposure to the virus, increased workload/working hours, fear of infecting their family and friends, lack of experience in managing the disease, perceived stigma, significant lifestyle changes, social discrimination and lack of personal protective equipment (PPE) [9–12]. The increased infection rate among healthcare workers is another important cause of such psychological impact [13].

During this crucial period, a more comprehensive understanding of the psychological burden among different groups of health service providers is essential so that appropriate psychological support could be provided and also strengthening mental healthcare could be done [14]. This cross-sectional study attempts to assess the psychological impact of COVID-19 and its associated variables among different healthcare workers in Nepal.

## Materials and methods

This study is a nationwide, web-based cross-sectional survey of psychological distress among health service providers during the COVID-19 pandemic in Nepal. Data for the survey were collected from 11 to 24 October 2020. The survey was filled by health professionals working in various institutes like hospitals, primary health centers, nursing homes, pharmacies, health posts and sub-health posts. Hospitals were teaching hospitals, district hospitals, regional hospitals, zonal hospitals and private hospitals. Medical professionals ranged from doctors, nurses, pharmacists, dentists, auxiliary health workers working in different departments ranging from intensive care units, wards, emergency departments, pharmacy shops, etc. For data collection purposes COVID-19 Peritraumatic Distress Index (CPDI) Questionnaire was used whose content validity was verified by Shanghai Mental Health Center [7]. As specified in the International Classification of Diseases, 11th Revision, apart from demographic data (age, gender,

religion, education, occupation, workload, availability of safety measures, nationality, ethnicity, and residence) the CPDI questionnaire includes relevant diagnostic guidelines for specific phobias and stress disorders and further inquiries about the frequency of anxiety, depression, cognitive change, avoidance, and compulsive behavior, physical symptoms and loss of social functioning in the past week, ranging from 0 to 96. Informed consent was taken from participants at the beginning of the questionnaire. For the survey, data were collected through an online Google form. Social media network was used to publish structured Google form with CPDI questions and forms were disseminated via email, messenger, Facebook group, Viber, etc. to the health care workers requesting them to participate in the survey and also to share the survey form to a wider audience. The eligible participants for the survey were physicians (including residents and fellows), advanced practice providers or registered nurses, and other service providers working at medical centers. Medical students were excluded from this survey as most of them usually do not enter the stage of clinical practice.

## Sample size

The minimum sample size required was 156. Sample size was determined using the formula:

$$N = [(z)^2 * p(1 - p)]/e^2;$$

where 'z' is 1.96 at 95% confidence interval, 'e' is margin of error at 5% and 'p' is prevalence rate of 11.5% from a recent study done in Nepal [15]. Adding 10% of the minimum sample as non- respondent, the desired sample size becomes 172.

## Exposure variables

Socioeconomic and demographic variables such as age (<30, 30–45, >45), Gender (male and female), Religion (non- Hinduism and Hinduism), Education (Diploma, bachelors or masters), employment (Doctor, nurse or other health care worker), Marital status (married, unmarried, widowed or divorced), Nationality (Nepali, non-Nepali), Ethnicity (Brahmin and Chhetri, Others), Residence (Province 1,2,3,4,5,6,7) were included in the survey questionnaire.

## Outcome variables

This study used the CPDI scale questionnaire with an additional socio-demographic questionnaire and the internal consistency of 24 CPDI variables was assessed by using Cronbach's α. Its internal reliability was found to be 0.905 indicating high internal consistency of the scale.

The 5- point Likert scoring system with scales ranging from never-0, occasionally-1, sometimes-2, often-3, always-4 was used. The total score thus calculated is classified as:- score between 0–28 is normal, 28 and 51 mild or moderate distress and $\geq 52$ severe distress.

## Statistical analysis

Data of the Survey was exported into Microsoft Excel-13. The data then imported, cleaned, categorized as appropriate, and analyzed using SPSS (Statistical Package for Social Science) version-22. For all the variables, a univariate analysis was performed to assess the distribution of each variable in frequency and the percentage to summarize categorical variables. Odds ratios of relevant predicting variables were estimated using logistic regression analysis which gives the relation between a set of predictor set X (exposure variable) and a dichotomous response variable Y(outcome variable). For ease, we specify the response to be Y = 0 or 1, with Y = 1 designating the occurrence of the event of interest. The outcome variable is No distress = 0 and distress = 1. The exposure variables were categorical.

### Research ethics

Before the survey, informed consent was obtained from all the respondents. The study was conducted following the protocol, approved by the ethics committee of Chitwan Medical College Teaching Hospital (letter-number- CMC-IRC/077/078-041).

## Results

A total of 257 health care workers participated in this survey. Three forms were incomplete so excluded and data from 254 participants were included in the analysis. The socioeconomic and demographic profile of responders is outlined in Table 1. In this study, the majority of respondents (85.4%) were less than 30 years old and the mean age of participants was 26.01 (±4.46) years. The male to female ratio is 1.01 with 50.04% male participants. The majority of participants were Hindu by religion (90.2%), Doctor by occupation (42.5%), completed bachelor's level or master's level (89.8%), and working in non-government hospitals (72%). Though most of the respondents work more than 4 days a week (71.1%) and more than or equal to 40 hrs per week (83.5%), almost two-thirds (63.8%) of these health care workers didn't receive any extra allowance. Approximately two-thirds of participants are residing in Bagmati province (61.8%).

Table 2 depicts the prevalence of every psychological component of the CPDI scale. More than two-third ($n = 222$, 87.4%) used to feel more nervous and anxious. Similarly, 74.8% of respondents ($n = 190$) felt insecure and bought a lot of masks, medications, sanitizers, gloves, and/or other home supplies. About half ($n = 120$, 47.2%) of the participants always felt sympathetic to COVID-19 patients and their families. Only approximately one third (n = 86, 33.9%) of the respondents believed the COVID-19 information from all sources without any validation. Approximately two-thirds ($n = 170$, 66.9%) didn't believe in negative news about COVID-19 and was not skeptical about the good news.

Table 3 demonstrates the distribution of severity of psychological distress by socioeconomic and demographic characteristics of Nepal. The frequency of mild to moderate distress among age groups <30 years, 30–45 years, and >45 years old were 104, 13, and 1 respectively whereas 14 severely distressed health service providers were below <30 years old. Female participants were having more distress (n = 80) compared to male participants (n = 55) which were statistically significant (p = 0.004). Additionally, participants who were doctors by profession experienced significantly more distress (n = 50, p = 0.001). Socioeconomic and demographic characteristics of participants like religion, education level, working hours, marital status, ethnicity, province of residence, and extra allowance were not significantly associated with distress level.

46.9% ($n = 119$) of the participants were not distressed while 46.5% (n = 118) were mild to moderate distressed and 6.7% (n = 17) were severely distressed due to COVID-19 pandemic (Fig 1).

Binary logistic regression analysis taking socio-demographic determinants of distress didn't show any significant association (**Table 1 in** S1 File).

## Discussion

A total of 254 health care workers from different provinces of the country participated in this study with a mean age of 26.01(± 4.46) years. The male to female ratio is 1.01 with 50.4% male participants. The findings of this study are consistent with another study conducted in Nepal, where 54.2% were male with a mean age of 27.8 years [16]. Though the survey was completed by a similar number of male and female participants with doctors and nurses being the largest two groups, the prevalence of distress among females was found to be higher which is

**Table 1. Socio-demographic profile of the health care workers (N = 254).**

| Socio-demographic variables | | Frequency | Percent |
|---|---|---|---|
| Age | <30 | 217 | 85.4 |
| | 30–45 | 36 | 14.2 |
| | >45 | 1 | .4 |
| | Mean ± SD | 26.01±4.46 | |
| Sex | Women | 126 | 49.6 |
| | Men | 128 | 50.4 |
| Religion | Non-Hinduism | 25 | 9.8 |
| | Hinduism | 229 | 90.2 |
| Education | Diploma | 26 | 10.2 |
| | Bachelor or master | 228 | 89.8 |
| Employment | Doctor | 108 | 42.5 |
| | Nurse | 61 | 24.0 |
| | Other HCW | 85 | 33.5 |
| Current Job | Government | 71 | 28.0 |
| | Non-Government | 183 | 72.0 |
| Institute category | Hospitals or higher center | 199 | 78.3 |
| | PHC, Health post, or others | 55 | 21.7 |
| Work in weeks per month | Less than 4 weeks per month | 48 | 18.9 |
| | 4 weeks per month | 206 | 81.1 |
| Duty HRS per week | Less than 40 hrs | 39 | 15.4 |
| | More than or equal 40 hrs | 212 | 83.5 |
| | Missing | 3 | 1.2 |
| Use of PPE | Complete set | 42 | 16.5 |
| | Incomplete | 209 | 82.3 |
| | Missing | 3 | 1.2 |
| Extra allowance | May be or Yes | 89 | 35.0 |
| | No | 162 | 63.8 |
| | Missing | 3 | 1.2 |
| Marital status | Married | 38 | 15.0 |
| | Unmarried | 214 | 84.3 |
| | Widowed or divorced | 2 | .8 |
| Nationality | Non-Nepali | 3 | 1.2 |
| | Nepali | 251 | 98.8 |
| Ethnicity | Brahmin and Chettri | 162 | 63.8 |
| | Others | 92 | 36.2 |
| Residence | Province 1 (Biratnagar as territorial capital) | 9 | 3.5 |
| | Province 2 (Janakpur as territorial capital) | 11 | 4.3 |
| | Province 3 (Bagmati) | 157 | 61.8 |
| | Province 4 (Gandaki) | 28 | 11.0 |
| | Province 5 (Butwal as territorial capital) | 34 | 13.4 |
| | Province 6 (Karnali) | 12 | 4.7 |
| | Province 7 (Sudurpaschim) | 3 | 1.2 |

NB: Nepal is yet to name all the provinces under the mandate of the new constitution and federal People's Republic

**Table 2. Presence of symptoms COVID-19 peri-traumatic distress (CPDI).**

| Questions | Never n(%) | Occasionally n(%) | Sometimes n(%) | Often n(%) | Always n(%) |
|---|---|---|---|---|---|
| Question 1: Compared to usual, I feel more nervous and anxious. | 32(12.6) | 45(17.7) | 111(43.7) | 50 (19.7) | 16(6.3) |
| Question 2: I feel insecure and bought a lot of masks, medications, sanitizers, gloves, and/or other home supplies. | 64(25.2) | 50(19.7) | 68(26.8) | 40 (15.7) | 32(12.6) |
| Question 3: I can't stop myself from imagining myself or my family being infected and feel terrified and anxious about it. | 39(15.4) | 54(21.3) | 66(26.0) | 55 (21.7) | 40(15.7) |
| Question 4: I feel helpless no matter what I do. | 100 (39.4) | 57(22.4) | 62(24.4) | 23(9.1) | 12(4.7) |
| Question 5: I feel sympathetic to COVID-19 patients and their families. | 7(2.8) | 21(8.3) | 39(15.4) | 67 (26.4) | 120 (47.2) |
| Question 6: I feel helpless and angry about people around me, governors, and media. | 41(16.1) | 41(16.1) | 75(29.5) | 51 (20.1) | 46(18.1) |
| Question 7: I am losing faith in the people around me. | 91(35.8) | 49(19.3) | 68(26.8) | 32 (12.6) | 14(5.5) |
| Question 8: I collect information about COVID-19 all day. Even if it's not necessary, I can't stop myself. | 73(28.7) | 63(24.8) | 58(22.8) | 30 (11.8) | 30(11.8) |
| Question 9: I will believe the COVID-19 information from all sources without any evaluation. | 168 (66.1) | 36(14.2) | 30(11.8) | 12(4.7) | 8(3.1) |
| Question 10: I would rather believe in negative news about COVID-19 and be skeptical about the good news. | 170 (66.9) | 33(13.0) | 31(12.2) | 7(2.8) | 13(5.1) |
| Question 11: I am constantly sharing news about COVID-19 (mostly negative news). | 172 (66.7) | 46(18.1) | 28(11.0) | 5(2.0) | 3(1.2) |
| Question 12: I avoid watching COVID-19 news since I am too scared to do so. | 140 (55.1) | 45(17.7) | 52(20.5) | 12(4.7) | 5(2.0) |
| Question 13: I am more irritable and have frequent conflicts with my family. | 140 (55.1) | 53(20.9) | 42(16.5) | 16(6.3) | 3(1.2) |
| Question 14: I feel tired and sometimes even exhausted. | 32(12.6) | 64(25.2) | 94(37.0) | 49 (19.3) | 15(5.9) |
| Question 15: When feelings anxious, my reactions are becoming sluggish. | 64(25.2) | 75(29.5) | 61(24.0) | 43 (16.9) | 11(4.3) |
| Question 16: I find it hard to concentrate. | 58(22.8) | 73(28.7) | 82(32.3) | 31 (12.2) | 10(3.9) |
| Question 17: I find it hard to make any decisions. | 71(28.0) | 80(31.5) | 70(27.6) | 22(8.7) | 11(4.3) |
| Question 18: During this COVID-19 period, I often feel dizzy or have back pain and chest distress. | 119 (46.9) | 57(22.4) | 55(21.7) | 17(6.7) | 6(2.4) |
| Question 19: During this COVID-19 period, I often feel stomach pain, bloating, and other stomach discomforts. | 129 (50.8) | 57(22.4) | 56(22.0) | 10(3.9) | 2(0.8) |
| Question 20: I feel uncomfortable when communicating with others. | 106 (41.7) | 61(24.0) | 56(22.0) | 23(9.1) | 8(3.1) |
| Question 21: Recently, I rarely talk to my family. | 145 (57.1) | 44(17.3) | 41(16.1) | 15(5.9) | 9(3.5) |
| Question 22: I have frequent awakening at night due to my dream about myself or my family being infected by COVID-19. | 184 (72.4) | 40(15.7) | 19(7.5) | 5(2.0) | 6(2.4) |
| Question 23: I have changes in my eating habits | 98(38.6) | 53(20.9) | 46(18.1) | 36 (14.2) | 21(8.3) |
| Question 24: I have constipation or frequent urination. | 173 (68.1) | 33(13.0) | 28(11.0) | 17(6.7) | 3(1.2) |

114 of the respondents (44.9%) would avoid watching COVID-19 news and a similar percentage of the respondents would be irritable and had a conflict with their family. More than half (*n* = 135, 53.1%) of the participants would feel dizzy or have back pain and chest distress and 49.2% of them would feel stomach pain, bloating, and other stomach discomforts. In addition to this, 58.3% would feel uncomfortable when communicating with others, 61.4% had changes in their eating habits and 31.9% had constipation or frequent urination.

**Table 3. Prevalence of CPDI by socioeconomic and demographic characteristics among HCWs in Nepal.**

| Socio-demographic Variables | | No distress (n) | Mild- moderate distress (n) | Severe distress (n) | p-value |
|---|---|---|---|---|---|
| Age | <30 | 99 | 104 | 14 | .575 |
| | 30–45 | 20 | 13 | 3 | |
| | >45 | 0 | 1 | 0 | |
| Sex | Women | 46 | 69 | 11 | .004 |
| | Men | 73 | 49 | 6 | |
| Religion | Non-Hinduism | 14 | 9 | 2 | .544 |
| | Hinduism | 105 | 109 | 15 | |
| Education | Diploma | 8 | 14 | 4 | .074 |
| | Bachelor or master | 111 | 104 | 13 | |
| Employment | Doctor | 58 | 46 | 4 | .001 |
| | Nurse | 16 | 36 | 9 | |
| | Other HCW | 45 | 36 | 4 | |
| Current Job | Government | 40 | 26 | 5 | .138 |
| | Non-Government | 79 | 92 | 12 | |
| Institute category | Hospitals or higher center | 89 | 97 | 13 | .376 |
| | PHC, Health post, or others | 30 | 21 | 4 | |
| Work in weeks per month | Less than 4 weeks per month | 19 | 27 | 2 | .293 |
| | 4 weeks per month | 100 | 91 | 15 | |
| Duty HRS per week | Less than 40 hrs | 20 | 18 | 1 | .490 |
| | More than or equal 40 hrs | 97 | 99 | 16 | |
| Use of PPE | Complete | 18 | 24 | 0 | .092 |
| | Incomplete | 99 | 93 | 17 | |
| Extra allowance | May be or Yes | 45 | 41 | 3 | .243 |
| | No | 72 | 76 | 14 | |
| Marital status | Married | 22 | 14 | 2 | .358 |
| | Unmarried | 97 | 102 | 15 | |
| | Widowed or divorced | 0 | 2 | 0 | |
| Nationality | Non-Nepali | 2 | 1 | 0 | .752 |
| | Nepali | 117 | 117 | 17 | |
| Ethnicity | Others | 36 | 48 | 8 | .156 |
| | Brahmin and Chhetri | 83 | 70 | 9 | |
| Residence | Province 1 (Biratnagar as territorial capital) | 7 | 2 | 0 | .232 |
| | Province 2 (Janakpur as territorial capital) | 4 | 5 | 2 | |
| | Province 3 (Bagmati) | 68 | 79 | 10 | |
| | Province 4 (Gandaki) | 18 | 10 | 0 | |
| | Province 5 (Butwal as territorial capital) | 15 | 16 | 3 | |
| | Province 6 (Karnali) | 6 | 5 | 1 | |
| | Province 7 (Sudurpaschim) | 1 | 1 | 1 | |

comparable with the outcome of a study conducted in Nepal [15] and Saudi Arabia [17]. Concern for family members and lack of proper knowledge regarding epidemics and public health emergencies may be the major cause for stress among females pointing towards the critical role of family and community support for mental health [18]. Higher workload and greater risk of direct exposure to COVID-19 patients have increased the vulnerability of females especially nurses for mental health [19].

Though it was a nationwide survey, we have a maximum number of participants from Bagmati province. The reason could be, this province includes the national capital city Kathmandu

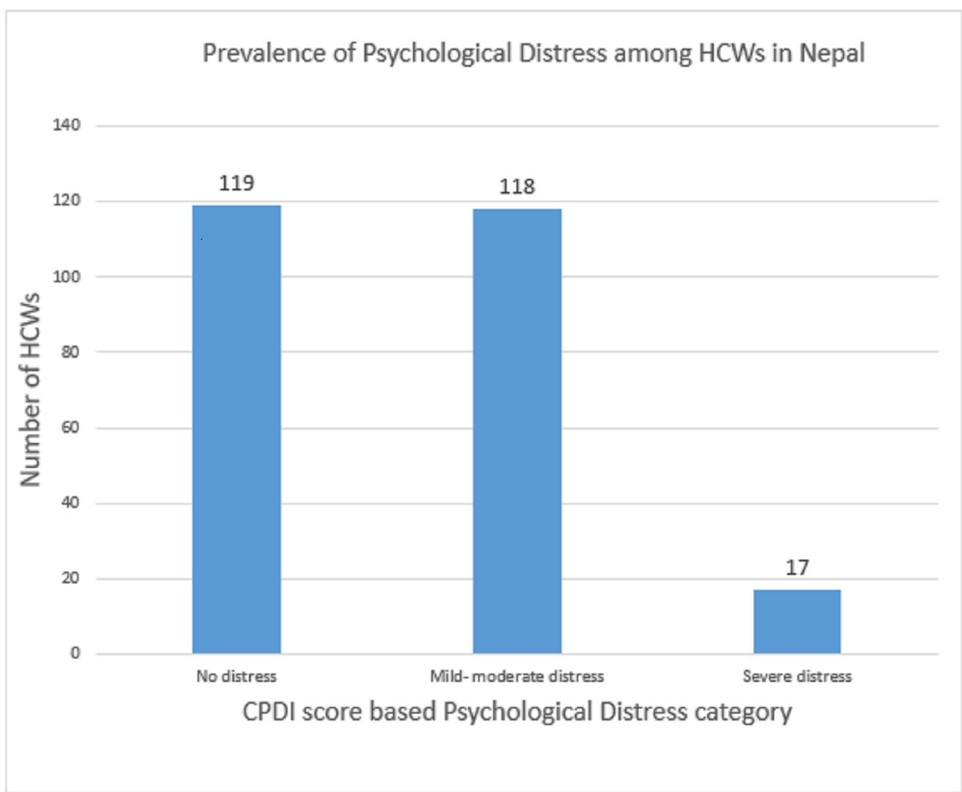

**Fig 1. Prevalence of psychological distress among HCWs in Nepal.**

and other major cities where comparatively a greater number of health professionals are supposed to be working. Most of the participants were educated till bachelor or higher level. The majority were working in non-government settings. This is expected as only a small proportion of all health forces are working under government and most of them are employed with non-government organizations [20].

A survey in China demonstrated that age, occupation, mass media report, and perception towards outbreak and public health emergencies bring significant variation in psychological distress among different individuals [21]. Many studies have shown that the risk of psychological problems is relatively more among health care workers than non-health workers as they are being exposed to patients with COVID-19 [22]. Psychological distress was found to be higher in doctors than in other HCWs in this study (p = 0.001). Doctors experience higher levels of mental stress during normal circumstances and health emergencies like COVID-19 exert additional pressure on doctors and the whole health care system [23].

The prevalence of mild distress was reported to be lower among health workers from China (36.5%) [24] and in Saudi Arabia (33.7%) [17] as compared to the findings of our study. In addition to this, the prevalence of mild-moderate distress (46.5%) and severe distress (6.7%) in this study was found to be higher as compared to a recent study conducted in Nepal among the general population, which showed that 11% of the participants had mild psychological distress while only 0.5% of them reported with severe distress [15]. This might be attributed greatly to the fact that healthcare workers are facing tremendous pressure from COVID-19 including a high risk of exposure to infection, inadequate protection due to shortage of healthcare resources, long duty hours, perceived stigma, lack of family contact, and the possibility of

family illness in addition to early and evolving nature of pandemic when the study was conducted [11, 25]. These factors can contribute to psychological problems in a substantial proportion of healthcare workers including depression, anxiety, insomnia, distress, obsessive-compulsive symptoms, and somatization symptoms [24, 26]. Notably, depression and post-traumatic stress symptoms might remain even after crises like the current pandemic are over [27, 28] and might as well surpass the consequences of the current pandemic itself [14].

The shortage of PPE wasn't statistically significant in our study, which might be due to adopting the participants with the current situation. However, several studies have reported this as a considerable source of distress among healthcare workers [25, 29, 30] and have specified the need to equip these frontline workers with adequate resources which can strengthen their overall work performance with better psychological outcomes [25, 31]. Lack of protective measures can create a sense of insecurity and thus imposes the healthcare workers to higher exposure to infections. Thus, these findings draw attention to the government of Nepal for providing adequate protective measures to lessen the escalating mental health burden among healthcare workers [30].

There is generally a higher risk of suicide among healthcare workers as compared to the general population [32] and COVID-19 has heightened this burden of suicide among healthcare workers [33]. There is no study relating to suicide rates in Nepal. However, a total of 1647 cases of the general population have committed suicide as of 27th June 2020 after the lockdown, which on average is 25% higher as compared to the pre-lockdown period [34]. Further studies are required to recuperate the magnitude of suicide among healthcare workers.

Expectedly, the findings from this study will help refine our understanding of the influence of the COVID-19 pandemic on psychological health among different groups of health service providers and highlight the need for appropriate implementation of plans that will help prevent and manage the distress among health service providers in the current pandemic. Moreover, for short term psychological problems like anxiety, depression, and insomnia, evidence-based psychosocial interventions and support are of utter necessity at the current stage [22].

## Limitation

Special consideration should be given while interpreting the data as the study had several limitations. In this online survey, a self-reported questionnaire was used and conducted in a nation where internet penetration is only 57% [35]. The use of cross-sectional data limits controls over unobserved heterogeneity among the respondents. It was a nationwide study where only a limited number of participants were involved. So, the sample may not necessarily be a good representation of the whole country and the generalizability of finding is limited. Also, there may be potential changes in distress with the progression of pandemic due to increasing number of cases and mortality. Majority of our participants were young and we did not evaluate the work experience of these young professionals. Lack of adequate work experience might have led to more distress. However, we could not determine these association of participant's working experience with distress due to lack of data on work experience. This might be explained by the fact that our survey was online web-based which were easier to fill by young medical professionals due to their technical expertise, easy access and widespread use compared to old medical professionals.

## Conclusions

This was a nationwide, web-based, cross-sectional study conducted to assess the psychological impact of COVID-19 and its associated factors among different healthcare workers in Nepal. More than half of health workers were categorized as having 'mild-to-severe distress' due to

the COVID-19 pandemic. Female participants and doctors were having significantly more distress. The findings from the present study may highlight the need for constructing and implementing appropriate plans and policies by relevant stakeholders that will help to mitigate the distress among health service providers in the current pandemic so that we can have an efficient frontline health workforce to tackle this worse situation.

## Supporting information

**S1 File. Questionnaire and supplement table.**
(DOCX)

## Acknowledgments

We would like to acknowledge Binaya Subedi, Pujan K.C. and Matrika Dhital for their assistance in Google form dissemination in social media and also to all the participants for their active response.

## Author Contributions

**Conceptualization:** Khagendra Kafle, Dhan Bahadur Shrestha, Abinash Baniya, Sandesh Lamichhane, Manoj Shahi, Bipana Gurung, Partiksha Tandan, Amrita Ghimire, Pravash Budhathoki.

**Data curation:** Khagendra Kafle, Dhan Bahadur Shrestha, Abinash Baniya, Sandesh Lamichhane, Manoj Shahi, Bipana Gurung, Partiksha Tandan, Amrita Ghimire, Pravash Budhathoki.

**Formal analysis:** Dhan Bahadur Shrestha.

**Investigation:** Khagendra Kafle, Dhan Bahadur Shrestha, Sandesh Lamichhane, Manoj Shahi, Bipana Gurung, Partiksha Tandan.

**Methodology:** Khagendra Kafle, Dhan Bahadur Shrestha, Abinash Baniya, Sandesh Lamichhane, Manoj Shahi, Bipana Gurung, Partiksha Tandan, Amrita Ghimire, Pravash Budhathoki.

**Project administration:** Khagendra Kafle, Dhan Bahadur Shrestha, Abinash Baniya, Sandesh Lamichhane, Manoj Shahi, Bipana Gurung, Partiksha Tandan, Amrita Ghimire, Pravash Budhathoki.

**Resources:** Khagendra Kafle, Abinash Baniya, Sandesh Lamichhane, Manoj Shahi, Bipana Gurung, Partiksha Tandan, Amrita Ghimire, Pravash Budhathoki.

**Software:** Dhan Bahadur Shrestha.

**Supervision:** Khagendra Kafle, Dhan Bahadur Shrestha.

**Validation:** Khagendra Kafle, Dhan Bahadur Shrestha.

**Writing – original draft:** Khagendra Kafle, Dhan Bahadur Shrestha, Abinash Baniya, Sandesh Lamichhane, Manoj Shahi, Bipana Gurung, Partiksha Tandan.

**Writing – review & editing:** Khagendra Kafle, Dhan Bahadur Shrestha, Abinash Baniya, Sandesh Lamichhane, Manoj Shahi, Bipana Gurung, Partiksha Tandan, Amrita Ghimire, Pravash Budhathoki.

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
