## [Decision Letter · Decision Letter 0]

14 Dec 2020

PONE-D-20-34279

Psychological Distress among Health Service Providers during COVID-19 Pandemic in Nepal

PLOS ONE

Dear Dr. Dhan Bahadur Shrestha

Thank you for submitting your manuscript to PLOS ONE. After careful consideration, we feel that it has merit but does not fully meet PLOS ONE’s publication criteria as it currently stands. Therefore, we invite you to submit a revised version of the manuscript that addresses the points raised during the review process.

Albeit nice and potentially interesting this study has some important limitations as highlighted by the two reviewers.

I hope Authors could be able to handle them in order to improve the quality of the manuscript and  the clarity of the message they want to give.

We look forward to receiving your revised manuscript.

Kind regards,

Simone Savastano

Academic Editor

PLOS ONE

Additional Editor Comments

Albeit nice and potentially interesting this study has some important limitations as highlighted by the two reviewers.

I hope Authors could be able to handle them in order to improve the quality of the manuscript and the clarity of the message they want to give.

2.The text in Figure 1 is hard to read. Please increase the size of the font.

Reviewers' comments:

Reviewer's Responses to Questions

**Comments to the Author**

1. Is the manuscript technically sound, and do the data support the conclusions?

Reviewer #1: Partly

Reviewer #2: Yes

2. Has the statistical analysis been performed appropriately and rigorously? 

Reviewer #1: I Don't Know

Reviewer #2: Yes

3. Have the authors made all data underlying the findings in their manuscript fully available?

Reviewer #1: Yes

Reviewer #2: Yes

4. Is the manuscript presented in an intelligible fashion and written in standard English?

Reviewer #1: Yes

Reviewer #2: Yes

5. Review Comments to the Author

Reviewer #1: The study raises several major comments and concerns.

1. The measurements of the prevalence of peritraumatic distress related to COVID-19 may be

influenced by the timing of the assessment with respect to the temporal evolution of the

pandemic. Specifically, the study was conducted in October 2020, i.e., several months after

the first outbreak. There is no mention of the potential changes and/or temporal evolution of

psychological distress over time and how it could have been influenced the main findings.

Do coping mechanisms develop over time or does the burden increase as the pandemic

progresses?

2. Another critical issue is that the setting of the study is unclear, i.e., “hospitals”, “higher

centers”, “health posts” are generic denominations. It is obvious that COVID-19-related

psychological distress is clearly influenced by the fact that healthcare workers have to deal

(or not) to patients with COVID-19. Was the study specifically focusing on workers who

had to care for this patient group or not? Importantly, the severity of COVID-19 is highly

variable (from asymptomatic cases to those in need of ICU admission). Thus, it is

conceivable that people who have to care to ICU cases are more scared and distressed than

those working in a general ward where patients with less severe disease are admitted.

3. How is it possible to claim that the pandemic has heightened the psychological distress

amongst health care service providers? While this is quite expected, the use of CPDI is

focused on COVID-19-related distress and broader implications (i.e., “the psychological

distress”) are unwarranted and not grounded in the study results.

4. The CPDI was developed in China; was it validated in the Nepalese population? Are

psychometric properties of the Nepalese version satisfactory?

5. Most participants were very young. Does working experience mitigate the psychological

distress elicited by COVID-19?

Minor points

1. Avoid mentioning statistical software in the “Abstract” section.

2. “ Gender” is a psychosocial construct; consider “sex” instead. Please also replace “female”

and “male” with “women” and “men”, respectively.

Reviewer #2: The study is well written and it focuses on an important aspect of the pandemic which has affected entire health care systems. However, there are several issues that have to be clarified. First of all, the vast majority of the population is less than 30 years old and this may not represent the overall population within the health care system. Second, the institute categories are not well specified especially when considering that the specific conditions in which employees work can affect their psychological distress regardless of the pandemic. Moreover, it's not clear whether all the participants worked directly with COVID-19 patients or not or how this distress level compares to pre-pandemic surveys of healthcare workers.

6. PLOS authors have the option to publish the peer review history of their article (what does this mean?). If published, this will include your full peer review and any attached files.

Reviewer #1: **Yes: **Benedetta Vanini

Reviewer #2: No

---

## [Author Response · Author response to Decision Letter 0]

29 Dec 2020

Joerg Heber,

Editor in Chief, PLOS ONE

Dear Dr. Heber,

Thank you for your interest in our manuscript: PONE-D-20-34279, Psychological Distress among Health Service Providers during COVID-19 Pandemic in Nepal. We are grateful to the editors for allowing us to revise our manuscript. We responded to the reviewer’s comments and concerns, and believe our manuscript is clearer and of greater quality as a result.

Below we repeat each of the comments from the referees in bold italics followed by our responses in plain text with yellow highlights.

Reply: We have amended in manuscript text.

2. The text in Figure 1 is hard to read. Please increase the size of the font.

 Reply: We have amended in manuscript text.

Reviewers' comments:

Reviewer's Responses to Questions

Comments to the Author

1. Is the manuscript technically sound, and do the data support the conclusions?

Reviewer #1: Partly

Reviewer #2: Yes

2. Has the statistical analysis been performed appropriately and rigorously?

Reviewer #1: I Don't Know

Reviewer #2: Yes

3. Have the authors made all data underlying the findings in their manuscript fully available?

Reviewer #1: Yes

Reviewer #2: Yes

4. Is the manuscript presented in an intelligible fashion and written in standard English?

Reviewer #1: Yes

Reviewer #2: Yes

5. Review Comments to the Author

Reviewer #1: The study raises several major comments and concerns.

1. The measurements of the prevalence of peritraumatic distress related to COVID-19 may be influenced by the timing of the assessment with respect to the temporal evolution of the pandemic. Specifically, the study was conducted in October 2020, i.e., several months after the first outbreak. There is no mention of the potential changes and/or temporal evolution of psychological distress over time and how it could have been influenced the main findings.

Do coping mechanisms develop over time or does the burden increase as the pandemic

progresses?

Reply: Distress severity is contrasted in discussion section. In similar survey in Nepalese residents using CDPI in early phase of pandemic showed distress in only 11%. While present survey showed significantly higher number of participants having distress, could be being this survey was only among health professional who has risk of contracting the infection and next may be due to evolution of pandemic. 

2. Another critical issue is that the setting of the study is unclear, i.e., “hospitals”, “higher

centers”, “health posts” are generic denominations. It is obvious that COVID-19-related

psychological distress is clearly influenced by the fact that healthcare workers have to deal (or not) to patients with COVID-19. Was the study specifically focusing on workers who had to care for this patient group or not? Importantly, the severity of COVID-19 is highly variable (from asymptomatic cases to those in need of ICU admission). Thus, it is

conceivable that people who have to care to ICU cases are more scared and distressed than those working in a general ward where patients with less severe disease are admitted.

Reply: This is detailed in methods sections in revision. Available data about institution and type of job is presented in table 1 of result section.

3. How is it possible to claim that the pandemic has heightened the psychological distress amongst health care service providers? While this is quite expected, the use of CPDI is focused on COVID-19-related distress and broader implications (i.e., “the psychological distress”) are unwarranted and not grounded in the study results.

Reply: Result of similar study regarding psychological distress in Nepalese community is contrasted in discussion. In prior study, may be due to early phase of pandemic distress level was relatively low in Nepal comparing with other countries, which were already in mid-later phase of pandemic. In present study, we found significantly higher proportion of health professionals with distress suggesting towards its relation with development of pandemic.

4. The CPDI was developed in China; was it validated in the Nepalese population? Are

psychometric properties of the Nepalese version satisfactory?

Reply: CPDI is widely under use due to its COVID-19 specific nature. In several studies carried out in Nepal and other countries are using it due to its specific nature. In our case we checked internal consistency of 24 CPDI variables using Cronbach's α. Its internal reliability was found to be 0.905 indicating high internal consistency of the scale.

5. Most participants were very young. Does working experience mitigate the psychological distress elicited by COVID-19?

Reply: We have explained in limitation section in discussion.

Minor points

1. Avoid mentioning statistical software in the “Abstract” section.

Reply: We have amended in manuscript text.

2. “Gender” is a psychosocial construct; consider “sex” instead. Please also replace “female” and “male” with “women” and “men”, respectively.

Reply: We have amended in manuscript text.

Reviewer #2: The study is well written and it focuses on an important aspect of the pandemic which has affected entire health care systems. However, there are several issues that have to be clarified. First of all, the vast majority of the population is less than 30 years old and this may not represent the overall population within the health care system. Second, the institute categories are not well specified especially when considering that the specific conditions in which employees work can affect their psychological distress regardless of the pandemic. Moreover, it's not clear whether all the participants worked directly with COVID-19 patients or not or how this distress level compares to pre-pandemic surveys of healthcare workers.

Reply: Amended, These issues discussed in method section and limitation section.

Thank you for your consideration. We look forward to hearing from you.

Sincerely,

Dhan Bahadur Shrestha

Mangalbare Hospital, Morang, Nepal

---

## [Decision Letter · Decision Letter 1]

27 Jan 2021

Psychological Distress among Health Service Providers during COVID-19 Pandemic in Nepal

PONE-D-20-34279R1

Dear Dr. Dhan Bahadur Shrestha

We’re pleased to inform you that your manuscript has been judged scientifically suitable for publication and will be formally accepted for publication once it meets all outstanding technical requirements.

Kind regards,

Simone Savastano

Academic Editor

PLOS ONE

Additional Editor Comments (optional):

Reviewers' comments:

Reviewer's Responses to Questions

**Comments to the Author**

1. If the authors have adequately addressed your comments raised in a previous round of review and you feel that this manuscript is now acceptable for publication, you may indicate that here to bypass the “Comments to the Author” section, enter your conflict of interest statement in the “Confidential to Editor” section, and submit your "Accept" recommendation.

Reviewer #1: All comments have been addressed

Reviewer #2: All comments have been addressed

2. Is the manuscript technically sound, and do the data support the conclusions?

Reviewer #1: (No Response)

Reviewer #2: Yes

3. Has the statistical analysis been performed appropriately and rigorously? 

Reviewer #1: (No Response)

Reviewer #2: Yes

4. Have the authors made all data underlying the findings in their manuscript fully available?

Reviewer #1: Yes

Reviewer #2: Yes

5. Is the manuscript presented in an intelligible fashion and written in standard English?

Reviewer #1: Yes

Reviewer #2: Yes

6. Review Comments to the Author

Reviewer #1: (No Response)

Reviewer #2: All the comments have been addressed except the comparison of the distress level with pre-pandemic surveys of healthcare workers. However, this could be an option for a future study in order to strenghten the results founded in this paper.

7. PLOS authors have the option to publish the peer review history of their article (what does this mean?). If published, this will include your full peer review and any attached files.

Reviewer #1: **Yes: **benedetta vanini

Reviewer #2: No

---

## [Editor Report · Acceptance letter]

29 Jan 2021

PONE-D-20-34279R1 

Psychological distress among health service providers during COVID-19 pandemic in nepal 

Dear Dr. Shrestha:

I'm pleased to inform you that your manuscript has been deemed suitable for publication in PLOS ONE. Congratulations! Your manuscript is now with our production department. 

Kind regards, 

on behalf of

Dr. Simone Savastano 

Academic Editor

PLOS ONE